# Lower Firmicutes abundance in gut microbiota associated with amyloid-β positivity in older adults in Japan as assessed by positron emission tomography

Narumi Kojima[1]*, Yosuke Osuka[1,2], Hiroyuki Sasai[1], Shoji Shinkai[1], Kenji Ishii[1], Noriyuki Kohda[3], Yodai Kobayashi[3], Hunkyung Kim[1]

1 Tokyo Metropolitan Institute for Geriatrics and Gerontology, Tokyo, Japan, 2 National Center for Geriatrics and Gerontology, Aichi, Japan, 3 Otsu Nutraceuticals Research Institute, Otsuka Pharmaceutical Co., Ltd., Shiga, Japan

* nkojima@tmig.or.jp

## Abstract

### Background

Alzheimer's disease, marked by amyloid-β accumulation, is a leading cause of dementia. Gut microbiota may influence its development by affecting inflammation or amyloid-β metabolism. However, this association is not well studied in older adults in Japan, whose characteristic diet may uniquely impact gut bacteria.

### Objective

To determine the association between the gut microbiota and positron emission tomography (PET)-determined brain amyloid-β positivity in community-dwelling older adults in Japan.

### Methods

This cross-sectional study investigated 136 participants aged 68–86 years from Tokyo. Brain amyloid-β was assessed using PET imaging, and gut microbiota were analyzed from fecal samples using 16S rRNA gene sequencing processed with the QIIME2 pipeline. Taxonomic composition was evaluated at both the phylum and genus levels; participants were classified into above- and below-median groups based on the relative abundance of each taxon. Binomial logistic regression adjusted for age, sex, and antibiotic use was conducted to examine the association between bacterial abundance and PET positivity. For genus-level analyses, p-values were further corrected for multiple comparisons. In addition, α diversity (Shannon, Simpson, Observed OTUs) and β diversity (PCoA based on unweighted UniFrac distances, PERMANOVA) were compared between PET-positive and PET-negative groups.

**Data availability statement:** The sequencing data reported in this paper are available in the DDBJ Sequence Read Archive (DRA) under BioProject accession number PRJDB37280. The data can be accessed at the following URLs: - DDBJ: https://ddbj.nig.ac.jp/public/ddbj_database/dra/fastq/DRA022/DRA022037/ - NCBI BioProject: https://www.ncbi.nlm.nih.gov/bioproject/PRJDB37280/ Other clinical datasets analyzed in this study (including amyloid PET results, demographic variables, and health-related parameters) contain potentially sensitive personal information. According to the policy of the Research Ethics Committee of the Tokyo Metropolitan Institute for Geriatrics and Gerontology, these data cannot be shared publicly. Data access requests can be directed to the Administrative Office of the Research Ethics Committee (E-mail: kenkyurinri@tmghig.jp), and will be considered for researchers who meet the criteria for access to confidential data.

**Funding:** Initials of the authors who received each award:SS, HK Grant numbers awarded to each author:Not applicable The full name of each funder:Otsu Nutraceuticals Research Institute, Otsuka Pharmaceutical Co., Ltd., Shiga, Japan. URL of each funder website:https://www.otsuka.co.jp/men-eki/labo/otsu-laboratory.html. Did the sponsors or funders play any role in the study design, data collection and analysis, decision to publish, or preparation of the manuscript:Yes.

**Competing interests:** This study was funded by Otsuka Pharmaceutical Co., Ltd., Shiga, Japan. SS and HK received research funding from Otsuka Pharmaceutical Co., Ltd. NoK and YK are employees of Otsuka Pharmaceutical Co., Ltd. and contributed mainly to the microbiome analyses and provided advice on the manuscript. The initial study design was discussed with representatives from Otsuka Pharmaceutical Co., Ltd. during the project planning stage. The funder had no role in the data collection, statistical analysis, or the decision to publish. The authors declare no additional competing interests. This does not alter our adherence to PLOS ONE policies on sharing data and materials.

## Results

Of the 136 participants, 34.6% were PET-positive for amyloid-β. Firmicutes showed a significant difference: 26.4% PET-positive in the above-median group vs. 42.6% in the below-median group (p for $\chi^2 = 0.047$). The binomial logistic regression analysis showed that lower Firmicutes abundance was significantly associated with an increased odds of PET positivity (odds ratio and confidence interval: 2.15 [1.03, 4.52]). At the genus level, no taxon remained significant after correction for multiple comparisons. No significant differences were observed in α or β diversity indices between groups.

## Conclusion

A lower abundance of Firmicutes may be associated with amyloid-β accumulation in the brain, linking the gut microbiota to Alzheimer's disease.

## Introduction

By 2040, one in three Japanese individuals aged 65 years or older is projected to develop dementia or mild cognitive impairment [1], posing a significant public health challenge. Among dementia pathologies, Alzheimer's disease (AD) is the most prevalent, characterized by amyloid-β accumulation in the brain. Recent research has highlighted the gut microbiota as a potential factor influencing AD, underscoring the role of gut bacteria in modulating neuroinflammation and amyloid-β metabolism. For instance, AD patients have shown altered gut microbial compositions, such as increased *Escherichia/Shigella* and decreased *Eubacterium rectale* [2], alongside reduced bacterial diversity [3]. A systematic review and meta-analysis further corroborated these findings, revealing decreased Firmicutes and increased Proteobacteria at the phylum level [4]. These discoveries suggest that gut microbiota may contribute to AD pathogenesis through complex interactions with systemic and brain physiology.

Despite these advances, the direct relationship between gut microbial composition and amyloid-β positivity in the brain remains underexplored. While gut bacteria are known to influence amyloid-β production [5] and degradation [6], existing studies have largely compared microbial profiles of diagnosed AD patients and healthy controls, leaving a gap in understanding how gut microbiota variations correlate with early markers of AD pathology, such as amyloid-β accumulation detected via positron emission tomography (PET). This gap in knowledge is critical, as elucidating these associations could refine our understanding of AD mechanisms and pave the way for microbiota-based interventions, including probiotic treatments shown in animal models to reduce amyloid-β deposition and preserve cognitive function [7].

Moreover, gut microbiota composition varies significantly across populations due to environmental and lifestyle factors, including diet and genetic background [8,9]. Japanese individuals, with their distinct dietary patterns rich in fermented foods and fiber, exhibit unique gut microbiota profiles compared to other populations [10].

However, studies linking these population-specific microbial characteristics to amyloid-β pathology are scarce. Investigating this relationship in Japanese older adults offers a unique opportunity to understand how dietary and microbial factors intersect with AD pathology.

This study aimed to address these gaps by analyzing the association between phylum-level gut microbial composition and brain amyloid-β positivity in community-dwelling older adults in Japan. By leveraging 16S rRNA sequencing to characterize gut microbiota and PET imaging to assess amyloid-β accumulation, we sought to explore the potential role of microbial phyla in AD pathology. The findings from this cross-sectional analysis may provide a foundation for future research, contributing to the development of targeted strategies for AD prevention and management through microbiota modulation.

## Materials and methods

### Study design and settings

This single-center cross-sectional study involved community-dwelling older adults who volunteered to participate. Participants were recruited from research-oriented comprehensive health examinations targeting residents of the Itabashi ward of Tokyo conducted by the Tokyo Metropolitan Institute for Geriatrics and Gerontology. This study was approved by the local ethics committee (TMIG-R21-19) in accord with the Helsinki Declaration of 1975.

### Study participants

Participants were selected from 1,517 men and women aged ≥ 65 years who underwent a health examination in September 2019. Invitations were sent via direct mail for brain imaging tests. After informed consent was obtained, patients were excluded for conditions such as epilepsy, depression, schizophrenia, dementia, Parkinson's disease, various chronic diseases, recent surgeries, or unsuitable magnetic resonance imaging (MRI) results. The participants underwent PET and fecal collection. The detailed inclusion and exclusion criteria were as follows:

*Inclusion criteria:* 1) Age: participants must be 65 years or older; 2) Participation in health examination: participants must have participated in the research-oriented comprehensive health examination conducted in September 2019; 3) Interest in brain imaging test results: participants must respond to direct mail expressing interest in obtaining their brain imaging test results; 4) Informed consent: participants must provide written informed consent after receiving an explanation of the advantages and disadvantages associated with participation.

*Exclusion criteria:* 1) Medical history: participants with a history of epilepsy, depression, schizophrenia, dementia, Parkinson's disease, drug hypersensitivity, chronic renal failure, chronic anemia, diabetes, gastrointestinal diseases (including gastrectomy, Crohn's disease, ulcerative colitis, irritable bowel syndrome, and stomach/liver/colon cancer); 2) Recent surgery: participants who underwent laparotomy within six months before stool collection; 3) MRI results: participants whose MRI results suggested dementia or other brain diseases; 4) Medical judgment: participants judged by a doctor to be unsuitable to participate.

### Fecal sample collection and processing

Fecal samples were collected using a guanidine thiocyanate solution (Feces Collection kit® FS-0007, Techno Suruga Lab, Shizuoka, Japan). After receiving instructions, each participant collected a fecal sample at home between two days prior to and on the day of the PET scan. The samples were stored at −20 °C or lesser temperature until DNA extraction.

### Calculation of the bacterial number composition

The collected stool samples were processed at the Genetic Engineering Research Group of Takara Bio CDM Center. Genomic DNA was isolated using a NucleoSpin 96 Soil kit (Macherey-Nagel, Düren, Germany) according to the manufacturer's instructions and purified using an Agencourt AMPure XP system (Beckman Coulter, Brea, CA, USA). The

                                                          

V3–V4 region of bacterial 16S rRNA genes was amplified via polymerase chain reaction (PCR) using a 16S (V3-V4) Metagenomic Library Construction Kit for NGS (Takara Bio) and a primer set of 341F (5'-TCGTCGGCAG CGTCAGATGT GTATAAGAGA CAGCCTACGG GNGGCWGCAG-3') and 806R (5'-GTCTCGTGGG CTCGGAGATG TGTATAAGAG ACAGGGACTA CHVGGGTWTC TAAT-3'). Thereafter, the PCR products were amplified using the Nextera XT index kit (Illumina) and the index sequences were added to the Illumina sequencer with a barcode sequence. Pooled libraries were sequenced using an Illumina MiSeq with a MiSeq v3 Reagent kit (Illumina) at Takara Bio Co. Ltd. (Shiga, Japan).

The sequence data obtained were processed using the standard QIIME2 pipeline (version 2020.2). Denoising was performed using the DADA2 plugin to generate amplicon sequence variants. Taxonomy assignment for each amplicon sequence variant was conducted using the silva-138–99-nb-classifier.qza, which can be downloaded from the Qiime2 website (QIIME 2, 2024) [11].

## PET scan procedure

PET scans were performed between July 15, 2020, and January 19, 2021, under one of the co-authors, physician KI. The diagnostic agent 18F-Flutemetamol (generic name: Vizamyl) for amyloid PET was manufactured in-house following Good Manufacturing Practice guidelines using an approved synthesizer FASTLab (GE Healthcare). A static PET scan was performed on a PET/computed tomography system (Discovery 710 or Discovery MI; GE Healthcare) for 20 min, starting 90 min after intravenous administration of 185 MBq of Vizamyl to the participants.

## Determination of positivity/negativity in amyloid PET

The obtained images were judged as positive/negative by a consensus of three physicians who had completed specialized training in the visual read of $^{18}$F-Flutemetamol PET. Amyloid PET images were evaluated following a validated method using image-pathology correlation in 68 autopsy brains in a phase III trial [12]. With this method, the presence of a "moderate to frequent stage" of amyloid pathology according to The Consortium to Establish a Registry for Alzheimer's Disease criteria [13] can be estimated with a sensitivity of 88% and a specificity of 88%.

In addition to the visual read, standardized uptake value ratio (SUVR) data were also obtained for each participant. For validation purposes, we conducted a concordance analysis between the binary classification based on the visual read and the SUVR-based classification using a conventional threshold of 0.58.

## Tests of cognitive function

The Japanese versions of the Mini-Mental State Examination (MMSE-J) [14] and Clinical Dementia Rating (CDR) [15] were administered to assess the participants' cognitive function. These tests were conducted for a mean of 117 days (standard deviation 36.7 days) before the date of the PET scan. Most family interviews in the CDR were conducted in person, whereas others were conducted via telephone.

## Statistical analysis

As a preliminary step, group comparisons of participant characteristics between amyloid PET-positive and PET-negative individuals were performed using independent-samples t-tests for continuous variables and chi-square tests (or Fisher's exact tests where appropriate) for categorical variables. For every bacterial phylum and genus, a χ-square test was performed for the association between the binary variable for the bacterial relative abundance (i.e., bacterial relative abundance is less or greater than the median among participants) and the PET result (positive/negative). Subsequently, for each phylum and genus, a binomial logistic regression analysis adjusted for age [16], sex, and use of antibiotics within 2 weeks was performed, with higher or lower than the median relative abundance of each phylum or genus as the independent variable and the amyloid PET result as the dependent variable, i.e., we used binary variables instead of continuous

proportion data as explanatory variables in logistic regression. We believe this approach ensures better interpretability and accounts for the compositional nature and skewness often present in proportion data. For genus-level analyses, p-values were additionally adjusted for multiple comparisons using the false discovery rate (FDR) method (Benjamini–Hochberg procedure). We did not apply multiple comparison correction to the six phyla tested because our primary hypothesis focused on the phylum Firmicutes, which has been associated with amyloid pathology in previous literature [3]. Other phyla were included as exploratory variables to provide a comprehensive picture, but the results should be interpreted with caution.

While the aim of this study was to examine the association between gut microbiota composition and amyloid PET positivity, considering multiple pathways, including those mediated by cognitive decline, since amyloid PET positivity likely precedes cognitive impairment, adjusting for cognitive status, such as MMSE scores, could overcontrol the model and create conceptual inconsistencies. To maintain focus on the direct association, adjustments were limited to sex, age, and antibiotic use, established confounders in gut microbiota research.

In addition to the primary analyses, we compared α and β diversity between PET-positive and PET-negative participants to explore potential differences in gut microbiota composition.

For α diversity, we assessed the Shannon Index, Simpson Index, and Observed OTUs. Each index was compared using a two-tailed independent t-test. Prior to the t-tests, Levene's test for equality of variances was conducted to confirm the assumption of equal variances.

For β diversity, we conducted a Principal Coordinates Analysis (PCoA) using unweighted UniFrac distance matrices. Statistical differences in community structure between the groups were assessed using PERMANOVA with 999 permutations. β diversity analyses were performed using QIIME2 (version 2020.2), and visualizations were generated in R (version 4.2.1) using the ggplot2 package (version 3.5.1).

Results were considered statistically significant at $P < 0.05$. IBM SPSS Statistics version 27 (Armonk, NY, USA) was used for α diversity statistical testing.

## Results

### Participant characteristics

Of the 150 individuals who responded to the recruitment process and fulfilled the inclusion and exclusion criteria, 136 completed both PET and gut microbiota tests. The final sample included 40 men (29.4%) and 96 women. The participants ranged in age from 68 to 86 years, with a mean of 79.2 years (SD = 4.0). The mean MMSE score was 28.3 (SD = 1.9), indicating generally preserved global cognitive function. However, 26.5% of participants had a CDR score of ≥ 0.5, suggestive of mild cognitive impairment.

Amyloid PET was positive in 47 participants (34.6%) and negative in 89 (65.4%). Comparisons between PET-positive and PET-negative groups revealed no significant differences in sex, age, or history of hypertension or diabetes mellitus. However, the PET-positive group had significantly fewer years of education (mean = 12.2 vs. 13.4 years, p = 0.006), lower MMSE scores (mean = 27.7 vs. 28.6, p = 0.012), and a higher proportion of individuals with CDR ≥ 0.5 (38.3% vs. 20.2%, p = 0.023) (Table 1).

To validate the reliability of the binary classification based on the visual read of amyloid PET images, we compared it with the SUVR-based classification using a cut-off value of 0.58. The concordance between the two methods was substantial, with a Cohen's kappa coefficient of 0.792.

### Comparison of bacterial composition between PET-positive and negative groups

Comparison of bacterial composition between PET-positive and negative groups showed a lower mean relative abundance of Firmicutes in the PET-positive group (relative abundance = 52.6%) compared to the PET-negative group (relative abundance = 56.8%) (Fig 1). The results of the χ-square test between the amyloid PET results and the binomial relative

**Table 1. Participant characteristics by amyloid PET status.**

| Variable | Total (N = 136) | PET-positive (N = 47) | PET-negative (N = 89) | p-value |
|---|---|---|---|---|
| Sex (Male, %) | 29.4 | 21.3 | 33.7 | 0.130[1] |
| Age, years (Mean±SD) | 79.2 ± 4.0 | 79.7 ± 4.0 | 78.9 ± 4.0 | 0.243[2] |
| Education years (Mean±SD) | 13.0 ± 2.5 | 12.2 ± 2.3 | 13.4 ± 2.5 | 0.006[2] |
| MMSE score (Mean±SD) | 28.3 ± 1.9 | 27.7 ± 2.1 | 28.6 ± 1.7 | 0.012[2] |
| History of diabetes mellitus (%) | 2.2 | 0.0 | 3.4 | 0.551[1+] |
| History of hypertension (%) | 48.5 | 48.9 | 48.3 | 0.945[1] |
| CDR ≥ 0.5 (%) | 26.5 | 38.3 | 20.2 | 0.023[1] |
| Amyloid PET-positive (%) | 34.6 | – | – | – |

Note:

SD: standard deviation, MMSE: Mini-Mental State Examination, CDR: Clinical Dementia Rating, PET: Positron Emission Tomography.

[1] p-value based on chi-square test. [2] p-value based on independent-samples t-test. + Fisher's exact test used due to small cell size.

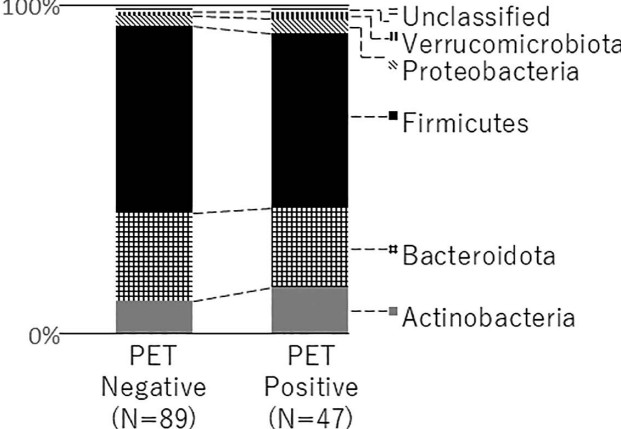

**Fig 1. Comparison between the amyloid PET-negative (left) and PET-positive groups (right) for the mean relative abundance of each phylum of the gut microbiota.** PET, positron emission tomography.

abundance of Firmicutes were statistically significant (p for $\chi^2$ = 0.047), with a higher percentage of participants having a relatively lower relative abundance of Firmicutes in the PET-positive group than in the PET-negative group. Similar statistically significant results were not observed for the remaining five phyla (Actinobacteria, Bacteroidota, Proteobacteria, Verrucomicrobiota, and unclassified phyla).

## Low Firmicutes abundance and amyloid PET positivity

At the phylum level, the odds ratio and 95% confidence interval, with adjustment for age, sex, and antibiotics use, of PET positivity in the group with fewer Firmicutes was 2.15 (1.03, 4.52) (p = 0.043) when the group with more Firmicutes was set as the reference (Table 2). No statistically significant odds ratios were observed for bacterial phyla other than Firmicutes (Table 2).

**Table 2. Odds ratios of amyloid PET positivity according to higher (reference group) or lower than the median relative abundance of intestinal bacteria at the phylum level.**

| Phylum | β | OR | 95% CI | p-value |
|---|---|---|---|---|
| Actinobacteria | −0.59 | 0.56 | 0.26-1.18 | 0.127 |
| Bacteroidota | 0.59 | 1.80 | 0.85-3.81 | 0.126 |
| Firmicutes | 0.77 | 2.15 | 1.03-4.52 | 0.043 |
| Proteobacteria | −0.31 | 0.74 | 0.36-1.52 | 0.407 |
| Verrucomicrobiota | 0.02 | 1.02 | 0.49-2.13 | 0.949 |
| Unclassified | 0.45 | 1.57 | 0.76-3.26 | 0.224 |

Adjusted for sex, age, antibiotics use. β: Regression coefficient, OR: Odds ratio, CI: Confidence interval. The reference group for each phylum is defined as participants with a relative abundance above the median value. Odds ratios (ORs) indicate the likelihood of amyloid PET positivity in participants with a relative abundance below the median compared to those in the reference group.

At the genus level, odds ratios for amyloid PET positivity were calculated for each genus using the same dichotomization approach (higher vs. lower than the median relative abundance) and adjusting for age, sex, and antibiotics use (Table 3). Several genera, including Clostridium, Ruthenibacterium, and Dorea, showed nominally significant associations (p<0.05). However, none of the associations remained statistically significant after controlling for multiple comparisons using the false discovery rate (FDR) correction.

### Comparison of gut microbiota diversity between PET-positive and negative groups

For α diversity, no significant differences were observed between PET-positive and PET-negative groups across all tested indices. The Shannon Index was 5.79±0.66 for the PET-positive group and 5.78±0.64 for the PET-negative group (t=−0.107, df=125, p=0.915; Levene's test: F=1.944, p=0.166). The Simpson Index showed similar results (means not significantly different; 0.957±0.037 v.s. 0.960±0.020, p=0.557; Levene's test: F=0.441, p=0.508), as did the Observed OTUs (197±56.4 v.s.188±62.4, p=0.391; Levene's test: F=1.174, p=0.281) (S1 Fig).

For β diversity, PCoA based on unweighted UniFrac distances revealed no distinct clustering between PET-positive and PET-negative groups. PERMANOVA analysis further confirmed that there were no significant differences in overall microbial community structure between the two groups (PERMANOVA p=0.661) (S2 Fig).

### Discussion

This cross-sectional study revealed a significant association between the Firmicutes composition and brain amyloid PET results in community-dwelling older adults in Japan. Even after adjusting for covariates, the odds ratio for PET positivity was significantly higher in patients with fewer Firmicutes than in those with more Firmicutes. At the genus level, several genera within Firmicutes and other phyla showed nominal associations with PET positivity; however, none remained statistically significant after FDR correction. These findings suggest that the observed phylum-level association may not be driven by a single genus with a strong effect, but rather by broader compositional features within the Firmicutes phylum. The consistency of our phylum-level results with previous studies in both Japanese and American populations [3]—despite differences in diet and gut microbiota composition—underscores the potential relevance of Firmicutes to amyloid-β accumulation.

### Potential mechanisms by which Firmicutes reduces amyloid-β accumulation

One possibility is that alterations in the gut microbiota, specifically a reduction in Firmicutes, may contribute to amyloid-β accumulation through mechanisms involving inflammation and metabolic dysregulation. Firmicutes, a major phylum in the

**Table 3. Odds ratios of amyloid PET positivity according to higher (reference group) or lower than the median relative abundance of intestinal bacteria at the genus level.**

| Genus | B | OR | 95% CI | p-value | FDR threshold | Sig. |
|---|---|---|---|---|---|---|
| Pauljensenia | −0.64 | 0.53 | 0.25-1.13 | 0.099 | 0.008 | |
| Bifidobacterium | 0.39 | 1.48 | 0.71-3.09 | 0.298 | 0.023 | |
| Rothia | 0.62 | 1.85 | 0.89-3.86 | 0.101 | 0.008 | |
| Collinsella | 0.24 | 1.27 | 0.61-2.65 | 0.520 | 0.036 | |
| Adlercreutzia | 0.56 | 1.75 | 0.84-3.63 | 0.136 | 0.014 | |
| Eggerthella | −0.04 | 0.96 | 0.46-2.01 | 0.922 | 0.048 | |
| Bacteroides | 0.05 | 1.05 | 0.50-2.21 | 0.896 | 0.045 | |
| Bacteroides_A[†] | 0.06 | 1.06 | 0.51-2.19 | 0.873 | 0.043 | |
| Bacteroides_B[†] | −0.48 | 0.62 | 0.30-1.29 | 0.200 | 0.019 | |
| Butyricimonas | 0.26 | 1.29 | 0.62-2.68 | 0.489 | 0.033 | |
| Odoribacter | 0.29 | 1.34 | 0.65-2.78 | 0.429 | 0.030 | |
| Alistipes | 0.60 | 1.83 | 0.88-3.82 | 0.108 | 0.010 | |
| Alistipes_A[†] | 0.61 | 1.84 | 0.88-3.84 | 0.106 | 0.009 | |
| Parabacteroides | −0.21 | 0.81 | 0.39-1.68 | 0.574 | 0.038 | |
| Bilophila | 0.68 | 1.98 | 0.95-4.15 | 0.070 | 0.004 | |
| Erysipelatoclostridium | −0.24 | 0.79 | 0.38-1.65 | 0.532 | 0.037 | |
| Absiella | 0.30 | 1.35 | 0.65-2.82 | 0.419 | 0.029 | |
| Holdemania | −0.03 | 0.97 | 0.47-2.01 | 0.939 | 0.048 | |
| Turicibacter | 0.45 | 1.56 | 0.75-3.26 | 0.235 | 0.021 | |
| Lactobacillus | 0.51 | 1.67 | 0.80-3.50 | 0.172 | 0.018 | |
| Streptococcus | 0.66 | 1.93 | 0.92-4.03 | 0.081 | 0.005 | |
| OEMS01[†] | 0.57 | 1.77 | 0.83-3.77 | 0.140 | 0.015 | |
| Christensenella | 0.44 | 1.56 | 0.75-3.24 | 0.235 | 0.022 | |
| Clostridium | 0.85 | 2.34 | 1.11-4.94 | 0.026 | 0.001 | |
| An114[†] | −0.05 | 0.95 | 0.46-1.99 | 0.898 | 0.046 | |
| Anaerotignum | −0.58 | 0.56 | 0.27-1.17 | 0.122 | 0.012 | |
| Acetatifactor | −0.71 | 0.49 | 0.24-1.03 | 0.060 | 0.003 | |
| Agathobacter | −0.09 | 0.91 | 0.44-1.89 | 0.801 | 0.042 | |
| Anaerostipes | −0.57 | 0.57 | 0.27-1.18 | 0.131 | 0.013 | |
| Blautia | −0.31 | 0.73 | 0.35-1.51 | 0.397 | 0.028 | |
| Blautia_A[†] | −0.46 | 0.63 | 0.30-1.31 | 0.216 | 0.020 | |
| CAG_81[†] | 0.03 | 1.03 | 0.49-2.16 | 0.943 | 0.049 | |
| Clostridium_M[†] | −0.39 | 0.68 | 0.32-1.43 | 0.306 | 0.024 | |
| Clostridium_Q[†] | 0.26 | 1.30 | 0.63-2.68 | 0.485 | 0.033 | |
| Dorea | −0.77 | 0.47 | 0.22-0.98 | 0.044 | 0.003 | |
| Eisenbergiella | 0.06 | 1.06 | 0.51-2.18 | 0.882 | 0.044 | |
| Eubacterium_E[†] | 0.08 | 1.08 | 0.52-2.23 | 0.835 | 0.043 | |
| Eubacterium_G[†] | −0.11 | 0.90 | 0.42-1.92 | 0.786 | 0.041 | |
| Eubacterium_I[†] | −0.29 | 0.75 | 0.36-1.55 | 0.436 | 0.031 | |
| Faecalicatena | −0.20 | 0.82 | 0.40-1.69 | 0.589 | 0.038 | |
| Fusicatenibacter | −0.60 | 0.55 | 0.26-1.16 | 0.115 | 0.011 | |
| GCA_900066575[†] | −0.36 | 0.70 | 0.34-1.45 | 0.340 | 0.026 | |
| KLE1615[†] | −0.36 | 0.70 | 0.34-1.46 | 0.343 | 0.027 | |
| Lachnospira | −0.58 | 0.56 | 0.27-1.18 | 0.127 | 0.013 | |
| Roseburia | −0.65 | 0.52 | 0.25-1.10 | 0.089 | 0.006 | |

*(Continued)*

**Table 3.** (Continued)

| Genus | B | OR | 95% CI | p-value | FDR threshold | Sig. |
|---|---|---|---|---|---|---|
| CAG_41† | −0.54 | 0.58 | 0.28-1.22 | 0.153 | 0.017 | |
| Acutalibacter | 0.55 | 1.74 | 0.83-3.62 | 0.142 | 0.016 | |
| Clostridium_A | 0.42 | 1.52 | 0.73-3.15 | 0.266 | 0.023 | |
| Ruminococcus_E† | 0.12 | 1.13 | 0.55-2.33 | 0.747 | 0.040 | |
| Agathobaculum | −0.51 | 0.60 | 0.29-1.25 | 0.172 | 0.018 | |
| Flavonifractor | 0.26 | 1.30 | 0.63-2.70 | 0.479 | 0.032 | |
| Lawsonibacter | 0.37 | 1.45 | 0.70-3.02 | 0.321 | 0.025 | |
| Oscillibacter | 0.05 | 1.05 | 0.51-2.17 | 0.898 | 0.047 | |
| Faecalibacterium | −0.25 | 0.78 | 0.38-1.61 | 0.500 | 0.034 | |
| Negativibacillus | 0.16 | 1.18 | 0.57-2.44 | 0.661 | 0.039 | |
| Ruminococcus_D† | −0.01 | 0.99 | 0.47-2.08 | 0.980 | 0.050 | |
| Ruthenibacterium | 0.79 | 2.20 | 1.05-4.64 | 0.038 | 0.002 | |
| UBA1191† | 0.62 | 1.87 | 0.90-3.89 | 0.096 | 0.007 | |
| Romboutsia | 0.25 | 1.28 | 0.62-2.66 | 0.507 | 0.035 | |
| Veillonella | 0.32 | 1.37 | 0.66-2.85 | 0.398 | 0.028 | |

Adjusted for sex, age, antibiotics use. β: Regression coefficient, OR: Odds ratio, CI: Confidence interval. The reference group for each genus is defined as participants with a relative abundance above the median value. Odds ratios (ORs) indicate the likelihood of amyloid PET positivity in participants with a relative abundance below the median compared to those in the reference group. † These are taxonomic labels from genome-based databases representing uncultured or unclassified genera. Genera with suffixes such as "_A", "_B", or "_Q" represent phylogenetically distinct clades within broader taxonomic groups, based on genome-based databases classifications. Genera detected in less than 50% of participants (i.e., those with a median relative abundance of zero) were excluded from the analysis.

gut microbiota, has been associated with anti-inflammatory properties. For example, a study on the effects of gut microbiota in a piglet model demonstrated that a predominance of Firmicutes was linked to lower levels of inflammation [17]. In contrast, a decrease in Firmicutes may disrupt this balance, promoting chronic low-grade inflammation, which is a recognized driver of insulin resistance [18]. Insulin resistance, in turn, is a key feature of type 2 diabetes and has been implicated in amyloid-β accumulation and neurodegeneration via its impact on insulin signaling in the brain [19]. This pathway suggests a potential link between reduced Firmicutes, systemic inflammation, and the metabolic conditions that promote amyloid-β pathology.

Furthermore, reduced levels of Firmicutes are often observed in individuals with type 2 diabetes [20] or obesity [21], both of which are associated with increased risk of Alzheimer's disease (AD). This evidence supports the hypothesis that a reduction in Firmicutes could act as a shared factor in metabolic dysfunction and neurodegenerative processes.

Furthermore, decreased Firmicutes decreases short-chain fatty acid (SCFA) production, which may contribute to increased amyloid-β. Along with Bacteroidetes, Firmicutes bacteria, especially those belonging to genera such as *Clostridium* and *Lactobacillus*, are important producers of SCFAs such as acetate, propionate, and butyrate through the fermentation of dietary substrates [22]. These SCFAs play crucial roles in gut health and host metabolism through anti-inflammatory and neuroprotective effects [23]. Therefore, the abundance and activity of Firmicutes in the gut microbiota can influence the production of SCFAs, which in turn can affect host health and metabolism. Reduced Firmicutes may diminish SCFA production, affect brain protection, and potentially enhance neurodegenerative processes, such as amyloid-β accumulation.

### Strengths and limitations of the study

The strength of this study is that it is the first to clarify the relationship between the gut microbiota and amyloid-β levels in the brain within a cohort in Japan, the super-aging society where the number of dementia patients is constantly

increasing. Though the level of amyloid-β in the brain should be closely related to cognitive decline, since cognitive function should be composed of extremely diverse elements, the influence of the gut environment may be very limited. In contrast, the accumulation of amyloid-β in the brain is a physiological parameter that may be relevant to cognitive function. We believe that it is significant that we were able to clarify for the first time the relationship between the intestinal microbiota and the accumulation of amyloid-β in the brain, which can be determined as an objective physiological value.

On the other hand, the limitations of this study are as follows: (i) the state of the intestinal microbiota can only be expressed in terms of the relative abundance of each bacterium; (ii) interpretation was limited to the phylum level for simplicity, although the classification of the gut bacteria itself was conducted at the genus level; (iii) the causal relationship is unknown because of the cross-sectional nature of the study; (iv) failure to take into account the ApoE genotype; and (v) not examining the association between gut bacteria and cognitive function. Concerning (i), the fact that intestinal microbiota can only be expressed in terms of the relative abundance of each bacterium is a general limitation of studies on intestinal bacteria. Analysis of absolute bacterial counts may lead to further elucidation of microbiota-dementia associations. Concerning (ii), the aim of the simple phylum level analysis was to avoid excessive complexity and to ensure clarity in the interpretation of the results. While this approach allowed us to identify a significant association between the Firmicutes phylum and amyloid PET positivity, it does not account for potential heterogeneity within the phylum. Further analyses at finer taxonomic levels, such as class or genus, could provide deeper insights into which specific bacterial groups within Firmicutes or other phyla might influence amyloid PET positivity. Addressing this limitation will be an important direction for future studies. Concerning (iii), the brain-gut correlation is bidirectional and the stress received by the brain may alter the intestinal microbiota [24]. In other words, it is difficult to assert from the results of this study alone that the amount of amyloid-β in the brain is causally affected by the presence of a particular bacterial species. Concerning (iv), ApoE4 is a known risk factor for the development of AD and has a significant impact on amyloid positivity rates [16]. Therefore, it is necessary to consider the ApoE genotype as a confounding factor in studies where amyloid PET is used as the dependent variable. However, because of the complexity of handling genetic information, genetic testing was not performed in this study. Concerning (v), although the association between gut microbiota and cognitive function is of interest, this study focused on amyloid PET positivity to maintain a clear and specific research objective. While data on cognitive function were available, including such analyses could have diverted focus from the primary aim. Future studies should explore the relationship between gut microbiota and cognitive status to provide a more comprehensive understanding of these interconnections.

## Conclusions

In community-dwelling older adults, the group with fewer Firmicutes phyla of intestinal bacteria had more brain amyloid PET-positive individuals than the group with more Firmicutes phyla. The result suggests that the composition of intestinal bacteria may be related to the concentration of amyloid-β in the brain and, ultimately, to the development of AD. Future research should focus on longitudinal studies to determine the causality of this relationship and consider the potential confounding effects of genetic factors, such as the ApoE genotype.

## Supporting information

**S1 Fig. Comparison of alpha diversity indices (Shannon, Simpson, Observed OTUs) between PET-positive and PET-negative groups.**
(TIF)

**S2 Fig. Principal coordinate analysis (PCoA) plot of β diversity based on unweighted UniFrac distances between amyloid PET-positive and -negative participants.**
(TIF)

## Acknowledgments

The authors have no acknowledgments to report.

## Author contributions

**Conceptualization:** Noriyuki Kohda, Yodai Kobayashi, Hunkyung Kim.

**Data curation:** Narumi Kojima.

**Formal analysis:** Narumi Kojima, Hiroyuki Sasai, Hunkyung Kim.

**Funding acquisition:** Shoji Shinkai, Noriyuki Kohda, Hunkyung Kim.

**Investigation:** Narumi Kojima, Yosuke Osuka, Hiroyuki Sasai, Kenji Ishii, Hunkyung Kim.

**Methodology:** Narumi Kojima, Yosuke Osuka, Hiroyuki Sasai, Shoji Shinkai, Kenji Ishii, Noriyuki Kohda, Yodai Kobayashi, Hunkyung Kim.

**Project administration:** Narumi Kojima, Shoji Shinkai, Kenji Ishii, Hunkyung Kim.

**Resources:** Kenji Ishii, Noriyuki Kohda, Yodai Kobayashi.

**Supervision:** Hiroyuki Sasai, Shoji Shinkai, Hunkyung Kim.

**Visualization:** Narumi Kojima.

**Writing – original draft:** Narumi Kojima.

**Writing – review & editing:** Yosuke Osuka, Hiroyuki Sasai, Kenji Ishii, Noriyuki Kohda, Yodai Kobayashi.

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
