## [Decision Letter · Decision Letter 0]

13 Jul 2025

Dear Dr. Kojima,

After careful consideration by 2 Reviewers and an Academic Editor, all of the critiques of the Reviewers must be addressed in detail in a revision to determine publication status. If you are prepared to undertake the work required, I would be pleased to reconsider my decision, but revision of the original submission without directly addressing the critiques of the Reviewers does not guarantee acceptance for publication in PLOS ONE. If the authors do not feel that the queries can be addressed, please consider submitting to another publication medium. A revised submission will be sent out for re-review. The authors are urged to have the manuscript given a hard copyedit for syntax and grammar. The authors also have to abide by PLOS ONE policy for data sharing as indicated by Reviewer #2.

We look forward to receiving your revised manuscript.

Kind regards,

Stephen D. Ginsberg, Ph.D.

Section Editor

PLOS ONE

2. Thank you for stating the following in the Competing Interests/Financial Disclosure * (delete as necessary) section: "Initials of the authors who received each award:SS, HK

Grant numbers awarded to each author:Not applicable

The full name of each funder:Otsu Nutraceuticals Research Institute, Otsuka Pharmaceutical Co., Ltd., Shiga, Japan.

URL of each funder website:https://www.otsuka.co.jp/men-eki/labo/otsu-laboratory.html

Did the sponsors or funders play any role in the study design, data collection and analysis, decision to publish, or preparation of the manuscript:Yes"

We note that you received funding from a commercial source: " Otsuka Pharmaceutical Co., Ltd., Shiga, Japan."

4. Please include captions for your Supporting Information files at the end of your manuscript, and update any in-text citations to match accordingly. Please see our Supporting Information guidelines for more information: http://journals.plos.org/plosone/s/supporting-information .

**Comments to the Author**

1. Is the manuscript technically sound, and do the data support the conclusions?

Reviewer #1: Yes

Reviewer #2: Partly

2. Has the statistical analysis been performed appropriately and rigorously?

Reviewer #1: No

Reviewer #2: No

3. Have the authors made all data underlying the findings in their manuscript fully available?

Reviewer #1: Yes

Reviewer #2: No

4. Is the manuscript presented in an intelligible fashion and written in standard English?

Reviewer #1: Yes

Reviewer #2: Yes

Reviewer #1: Existing studies have largely compared microbial profiles of diagnosed AD patients and healthy controls, leaving a gap in understanding how gut microbiota variations correlate with early markers of AD pathology such as amyloid. This study address this gap in sample from Japanese population with their distinct dietary patterns rich in fermented foods and fiber, exhibit unique gut microbiota profiles compared to other populations. This study is of cross-sectional. The results are interesting to the readership of this journal and the novel in research questions and findings, but is relatively weak.

Comments:

1) Was 20 minute scan of PET static?

2) The PET read was only visual and binary (with strict traing and guideline). The Consotium to Establish a Registry for AD is with 88% sensitivity and 88% specificity. This can be potential soruce of misdiagnosis.

3) The alpha and beta diversity with either t-test of PCoA test is not well described. More details are needed.

4) CDR>=0.5 is indicative of some cognition impairment. The percentage amyloid positivity 38.3% vs 20.2% could be biologically meaningful and statistially significant. This should be examined. The same is true for each item listed in Table 1

5) there are 6 phyla, and the binary logistic regression result significance may need to correct for mulitple comparison (6 of them). If not, justification is needed (firmicutes is reported in some other studies in other populations?)

Reviewer #2: Manuscript investigates microbiome changes associated with amyloid status using 16S rRNA gene sequencing. While the topic is of journal relevancy, the study suffers from two major issues that must be addressed before further consideration for thorough review.

1. Lack of genus-level analysis despite using 16S Amplicon gene sequencing.

The manuscript misses an opportunity to explore known or novel associations at a more informative taxonomic resolution. Given that standard 16S pipelines (e.g., SILVA, Greengenes2) support genus-level classification, this omission appears unjustified. The authors should present differentially abundant genera. Genus-level insights are essential for identifying biologically relevant taxa and comparing with existing literature.

Alpha (Shannon but also Simpson, Observed) and beta diversity plots should also be presented irrespective if they are negative or positive results.

2. Missing public data availability — violates PLOS ONE policy

The manuscript does not provide access to any raw or processed microbiome data, such as:

FASTQ files from the sequencing run

Feature/ASV tables or taxonomic annotations

This is a clear violation of PLOS ONE's data availability policy:

“Authors must follow standards and practice for data deposition in publicly available resources including those created for gene sequences, microarray expression, structural studies, and similar kinds of data.”

The authors must deposit their data in a public repository such as NCBI, ENA, or any other and provide accession numbers in the revised manuscript. Please check journal guidelines for clinical data.

**Do you want your identity to be public for this peer review?**  For information about this choice, including consent withdrawal, please see our Privacy Policy

Reviewer #1: **Yes: ** Kewei Chen

Reviewer #2: **Yes: ** Jaspreet Singh Saini

---

## [Author Response · Author response to Decision Letter 1]

27 Aug 2025

Reviewer comment (R1-1):

“Was 20 minute scan of PET static?”

Response:

Yes, the PET scan used in this study was performed as a static scan. To clarify this point, we have revised the Methods section and now explicitly state that a static PET scan was conducted. The revised sentence reads as follows.

[Changes highlighted in red in the revised manuscript. Line 146]

"A static PET scan was performed on a PET/computed tomography system (Discovery 710 or Discovery MI; GE Healthcare) for 20 min, starting 90 min after intravenous administration of 185 MBq of Vizamyl to the participants."

Reviewer comment (R1-2):

“The PET read was only visual and binary (with strict traing and guideline). The Consotium to Establish a Registry for AD is with 88% sensitivity and 88% specificity. This can be potential soruce of misdiagnosis.”

Response:

We thank the reviewer for this insightful comment. In addition to the visual assessment, we have confirmed that quantitative data based on SUVR (Standard Uptake Value Ratio) was available in our dataset. To evaluate the agreement between the visual read and SUVR-based classification (cut-off = 0.58), we conducted an analysis using Cohen’s kappa coefficient. The result showed κ = 0.792, indicating substantial agreement between the two methods. This finding supports the validity of using the visual read in the current analysis. We have now included this analysis and the kappa value in the revised manuscript (Methods and Results sections).

[Changes highlighted in red in the revised manuscript. Line 158]

“In addition to the visual read, standardized uptake value ratio (SUVR) data were also obtained for each participant. For validation purposes, we conducted a concordance analysis between the binary classification based on the visual read and the SUVR-based classification using a conventional threshold of 0.58.”

[Changes highlighted in red in the revised manuscript. Line 223]

“To validate the reliability of the binary classification based on the visual read of amyloid PET images, we compared it with the SUVR-based classification using a cut-off value of 0.58. The concordance between the two methods was substantial, with a Cohen’s kappa coefficient of 0.792.”

Reviewer comment (R1-3):

“The alpha and beta diversity with either t-test of PCoA test is not well described. More details are needed.”

Response:

Thank you very much for this important comment. In response, we have revised both the Materials and Methods and Results sections to provide more comprehensive descriptions of our diversity analyses.

In the Materials and Methods, we now explicitly specify:

That we evaluated three indices for α diversity: Shannon Index, Simpson Index, and Observed OTUs.

That each index was analyzed using a two-tailed independent t-test.

That Levene’s test was used to assess the assumption of equal variances prior to each t-test, and that all results supported the use of the equal variance assumption.

That for β diversity, we employed unweighted UniFrac distance metrics and visualized the results using PCoA.

That PERMANOVA was performed with 999 permutations using QIIME2 (version 2020.2), and PCoA visualizations were created with R (version 4.2.1) and the ggplot2 package (version 3.5.1).

In the Results, we have now included:

Mean values and p-values for each α diversity index, along with Levene’s test statistics.

Clear reporting of the statistical outcome of the PERMANOVA test.

A summary statement that no significant differences were observed in either α or β diversity between amyloid-positive and amyloid-negative participants.

We believe these additions greatly improve the transparency and reproducibility of our analytical approach.

[Changes highlighted in red in the revised manuscript. Line 34]

“In addition, α diversity (Shannon, Simpson, Observed OTUs) and β diversity (PCoA based on unweighted UniFrac distances, PERMANOVA) were compared between PET-positive and PET-negative groups.”

[Changes highlighted in red in the revised manuscript. Line 42]

“No significant differences were observed in α or β diversity indices between groups.”

[Changes highlighted in red in the revised manuscript. Line 195]

“In addition to the primary analyses, we compared α and β diversity between PET-positive and PET-negative participants to explore potential differences in gut microbiota composition.

For α diversity, we assessed the Shannon Index, Simpson Index, and Observed OTUs. Each index was compared using a two-tailed independent t-test. Prior to the t-tests, Levene’s test for equality of variances was conducted to confirm the assumption of equal variances.

For β diversity, we conducted a Principal Coordinates Analysis (PCoA) using unweighted UniFrac distance matrices. Statistical differences in community structure between the groups were assessed using PERMANOVA with 999 permutations. β diversity analyses were performed using QIIME2 (version 2020.2), and visualizations were generated in R (version 4.2.1) using the ggplot2 package (version 3.5.1).

Results were considered statistically significant at P < 0.05. IBM SPSS Statistics version 27 (Armonk, NY, USA) was used for α diversity statistical testing.”

[Changes highlighted in red in the revised manuscript. Line 286]

“For α diversity, no significant differences were observed between PET-positive and PET-negative groups across all tested indices. The Shannon Index was 5.79 ± 0.66 for the PET-positive group and 5.78 ± 0.64 for the PET-negative group (t = -0.107, df = 125, p = 0.915; Levene’s test: F = 1.944, p = 0.166). The Simpson Index showed similar results (means not significantly different; 0.957 ± 0.037 v.s. 0.960 ± 0.020, p = 0.557; Levene’s test: F = 0.441, p = 0.508), as did the Observed OTUs (197 ± 56.4 v.s.188 ± 62.4, p = 0.391; Levene’s test: F = 1.174, p = 0.281) (S1 Fig).

For β diversity, PCoA based on unweighted UniFrac distances revealed no distinct clustering between PET-positive and PET-negative groups. PERMANOVA analysis further confirmed that there were no significant differences in overall microbial community structure between the two groups (PERMANOVA p = 0.661) (S2 Fig).”

Reviewer comment (R1-4):

“CDR>=0.5 is indicative of some cognition impairment. The percentage amyloid positivity 38.3% vs 20.2% could be biologically meaningful and statistially significant. This should be examined. The same is true for each item listed in Table 1”

Response:

Thank you very much for your insightful suggestion. Although Table 1 was originally presented for descriptive purposes only, we agree that a statistical comparison between amyloid PET-positive and PET-negative groups can provide additional interpretative value.

Therefore, we conducted appropriate statistical tests (independent t-tests for continuous variables and chi-square tests for categorical variables) to examine differences between the two groups across the characteristics listed in Table 1.

The results have now been added to Table 1 (including p-values), and a brief summary has been included in the Results section. These analyses are presented for exploratory purposes only and do not alter the primary outcomes of the study.

[Changes highlighted in red in the revised manuscript. Line 171]

“As a preliminary step, group comparisons of participant characteristics between amyloid PET-positive and PET-negative individuals were performed using independent-samples t-tests for continuous variables and chi-square tests (or Fisher’s exact tests where appropriate) for categorical variables.”

[Changes highlighted in red in the revised manuscript. Line 212]

“The final sample included 40 men (29.4%) and 96 women. The participants ranged in age from 68 to 86 years, with a mean of 79.2 years (SD = 4.0). The mean MMSE score was 28.3 (SD = 1.9), indicating generally preserved global cognitive function. However, 26.5% of participants had a CDR score of ≥ 0.5, suggestive of mild cognitive impairment.

Amyloid PET was positive in 47 participants (34.6%) and negative in 89 (65.4%). Comparisons between PET-positive and PET-negative groups revealed no significant differences in sex, age, or history of hypertension or diabetes mellitus. However, the PET-positive group had significantly fewer years of education (mean = 12.2 vs. 13.4 years, p = 0.006), lower MMSE scores (mean = 27.7 vs. 28.6, p = 0.012), and a higher proportion of individuals with CDR ≥ 0.5 (38.3% vs. 20.2%, p = 0.023) (Table 1).”

Reviewer comment (R1-5):

“there are 6 phyla, and the binary logistic regression result significance may need to correct for mulitple comparison (6 of them). If not, justification is needed (firmicutes is reported in some other studies in other populations?)”

Response:

We appreciate the reviewer’s thoughtful comment regarding the potential need for multiple comparison correction in the phylum-level analyses. In our study, the primary hypothesis concerned the association between the Firmicutes phylum and amyloid PET positivity, as suggested by previous literature [3]. Therefore, Firmicutes was treated as the primary phylum of interest, and the other five phyla were included as exploratory variables to provide a comprehensive overview. In line with this hypothesis-driven approach, we did not apply multiple comparison correction to the phylum-level analyses. We have now clarified this rationale in the Statistical analysis section of the Methods. We believe this approach appropriately balances hypothesis testing and exploratory analysis, while making the rationale for the statistical treatment transparent to the reader.

[Changes highlighted in red in the revised manuscript. Line 185]

“We did not apply multiple comparison correction to the six phyla tested because our primary hypothesis focused on the phylum Firmicutes, which has been associated with amyloid pathology in previous literature [3]. Other phyla were included as exploratory variables to provide a comprehensive picture, but the results should be interpreted with caution.”

Reviewer comment (R2-1-1):

“Lack of genus-level analysis despite using 16S Amplicon gene sequencing.”

Response:

We thank the reviewer for the suggestion to explore the association between amyloid PET positivity and bacterial taxa at the genus level. In response, we conducted additional genus-level analyses. For each genus, we categorized participants into higher or lower than the median relative abundance groups, performed binomial logistic regression adjusted for age, sex, and recent antibiotic use, and applied false discovery rate correction (Benjamini–Hochberg procedure) to account for multiple comparisons.

The results of these analyses have been added as Table 3 in the Results section. After FDR adjustment, no genera showed statistically significant associations with amyloid PET positivity. The Methods (Statistical analysis section), Results, and Discussion have been updated to describe the analytical approach, present the findings, and discuss the implications of the absence of significant genus-level associations.

We believe these additions provide a more comprehensive understanding of the gut microbiota–amyloid PET relationship and transparently convey both significant and non-significant findings.

[Changes highlighted in red in the revised manuscript. Line 30]

“Taxonomic composition was evaluated at both the phylum and genus levels; participants were classified into above- and below-median groups based on the relative abundance of each taxon.”

[Changes highlighted in red in the revised manuscript. Line 33]

“For genus-level analyses, p-values were further corrected for multiple comparisons. In addition, α diversity (Shannon, Simpson, Observed OTUs) and β diversity (PCoA based on unweighted UniFrac distances, PERMANOVA) were compared between PET-positive and PET-negative groups.”

[Changes highlighted in red in the revised manuscript. Line 41]

“At the genus level, no taxon remained significant after correction for multiple comparisons. No significant differences were observed in α or β diversity indices between groups.”

[Changes highlighted in red in the revised manuscript. Line 174]

“For every bacterial phylum and genus, a χ-square test was performed for the association between the binary variable for the bacterial relative abundance (i.e., bacterial relative abundance is less or greater than the median among participants) and the PET result (positive/negative). Subsequently, for each phylum and genus, a binomial logistic regression analysis adjusted for age [16], sex, and use of antibiotics within 2 weeks was performed, with higher or lower than the median relative abundance of each phylum or genus as the independent variable and the amyloid PET result as the dependent variable, i.e we used binary variables instead of continuous proportion data as explanatory variables in logistic regression.”

[Changes highlighted in red in the revised manuscript. Line 183]

“For genus-level analyses, p-values were additionally adjusted for multiple comparisons using the false discovery rate (FDR) method (Benjamini–Hochberg procedure). We did not apply multiple comparison correction to the six phyla tested because our primary hypothesis focused on the phylum Firmicutes, which has been associated with amyloid pathology in previous literature [3]. Other phyla were included as exploratory variables to provide a comprehensive picture, but the results should be interpreted with caution.”

[Changes highlighted in red in the revised manuscript. Line 265]

“At the genus level, odds ratios for amyloid PET positivity were calculated for each genus using the same dichotomization approach (higher vs. lower than the median relative abundance) and adjusting for age, sex, and antibiotics use (Table 3). Several genera, including Clostridium, Ruthenibacterium, and Dorea, showed nominally significant associations (p < 0.05). However, none of the associations remained statistically significant after controlling for multiple comparisons using the false discovery rate (FDR) correction.”

[Changes highlighted in red in the revised manuscript. Line 272]

“Table 3. Odds ratios of amyloid PET positivity according to higher (reference group) or lower than the median relative abundance of intestinal bacteria at the genus level”

[Changes highlighted in red in the revised manuscript. Line 275]

“Adjusted for sex, age, antibiotics use. β: Regression coefficient, OR: Odds ratio, CI: Confidence interval. The reference group for each genus is defined as participants with a relative abundance above the median value. Odds ratios (ORs) indicate the likelihood of amyloid PET positivity in participants with a relative abundance below the median compared to those in the reference group. † These are taxonomic labels from genome-based databases representing uncultured or unclassified genera. Genera with suffixes such as "_A", "_B", or "_Q" represent phylogenetically distinct clades within broader taxonomic groups, based on genome-based databases classifications. Genera detected in less than 50% of participants (i.e., those with a median relative abundance of zero) were excluded from the analysis.”

[Changes highlighted in red in the revised manuscript. Line 302]

“At the genus level, several genera within Firmicutes and other phyla showed nominal associations with PET positivity; however, none remained statistically significant after FDR correction. These findings suggest that the observed phylum-level association may not be driven by a single genus with a strong effect, but rather by broader compositional features within the Firmicutes phylum. The consistency of our phylum-level results with previous studies in both Japanese and American populations [3]—despite differences in diet and gut microbiota composition—underscores the potential relevance of Firmicutes to amyloid-β accumulation.”

Reviewer comment (R2-1-2):

“Alpha (Shannon but also Simpson, Observed) and beta diversity plots should also be presented irrespective if they are negative or positive results.”

Response:

We appreciate the Reviewer’s suggestion. In response, we have now added figures to illustrate both α and β diversity results. Specifically, Supplementary Figure 1 presents boxplots of the Shannon, Simpson, and Observed OTUs

---

## [Decision Letter · Decision Letter 1]

4 Sep 2025

Lower Firmicutes abundance in gut microbiota associated with amyloid-β positivity in older adults in Japan as assessed by positron emission tomography

PONE-D-25-11508R1

Dear Dr. Kojima,

We’re pleased to inform you that your manuscript has been judged scientifically suitable for publication and will be formally accepted for publication once it meets all outstanding technical requirements.

Kind regards,

Stephen D. Ginsberg, Ph.D.

Section Editor

PLOS ONE

**Comments to the Author**

Reviewer #1: All comments have been addressed

2. Is the manuscript technically sound, and do the data support the conclusions?

Reviewer #1: Yes

3. Has the statistical analysis been performed appropriately and rigorously?

Reviewer #1: Yes

4. Have the authors made all data underlying the findings in their manuscript fully available?

Reviewer #1: No

5. Is the manuscript presented in an intelligible fashion and written in standard English?

Reviewer #1: Yes

Reviewer #1: responses are well done. Especially in the careful performance of the new statistical analysis, method description and results reporting

**Do you want your identity to be public for this peer review?** For information about this choice, including consent withdrawal, please see our Privacy Policy

Reviewer #1: **Yes: ** Dr Kewei Chen

---

## [Editor Report · Acceptance letter]

PONE-D-25-11508R1

PLOS ONE

Dear Dr. Kojima,

I'm pleased to inform you that your manuscript has been deemed suitable for publication in PLOS ONE. Congratulations! Your manuscript is now being handed over to our production team.

Kind regards,

on behalf of

Dr. Stephen D. Ginsberg

Section Editor

PLOS ONE